# Interpreted Petri Nets Applied to Autonomous Components within Electric Power Systems

Iwona Grobelna * and Paweł Szcześniak 

Institute of Automatic Control, Electronics and Electrical Engineering, University of Zielona Góra, 65-516 Zielona Góra, Poland; p.szczesniak@iee.uz.zgora.pl
* Correspondence: i.grobelna@iee.uz.zgora.pl

**Abstract:** In this article, interpreted Petri nets are applied to the area of power and energy systems. These kinds of nets, equipped with input and output signals for communication with the environment, have so far proved to be useful in the specification of control systems and cyber–physical systems (in particular, the control part), but they have not been used in power systems themselves. Here, interpreted Petri nets are applied to the specification of autonomous parts within power and energy systems. An electric energy storage (EES) system is presented as an application system for the provision of a system service for stabilizing the power of renewable energy sources (RES) or highly variable loads. The control algorithm for the EES is formally written as an interpreted Petri net, allowing it to benefit from existing analysis and verification methods. In particular, essential properties of such specifications can be checked, including, e.g., liveness, safety, reversibility, and determinism. This enables early detection of possible structural errors. The results indicate that interpreted Petri nets can be successfully used to model and analyze autonomous control components within power energy systems.

**Keywords:** control system; Petri net; specification; power and energy system



## 1. Introduction

Petri nets [1] are a general modeling formalism introduced for discrete event systems [2], including automatic control, with simple structures and easy creation rules [3]. Their basic elements include places and transitions connected alternately with each other via arcs and tokens that indicate the current state of the system. They are widely supported by existing analysis and verification methods [4], as well as editing tools [5], and are currently commonly used in many areas, such as manufacturing systems [6], freight logistics, and transportation systems [7]. There are also some approaches for their use in other domains such as energy, industrial electronics, or power systems.

A survey paper [8] summarized the application of Petri nets in the area of power systems up to the year 2006, listing some papers focusing on fault diagnosis, power system restoration, distribution network reconfiguration, unit commitment, power network topology analysis, reliability analysis, protective relay modeling, and hybrid power systems. Since then, some other approaches have appeared. Let us point out some of the most interesting ones. Load sharing control in distributed generation applications was supported by Petri nets in [9]. Hybrid Petri nets were used to analyze contingencies in power systems in [10]. In [11], a Petri net model with inhibitor arcs was applied to describe a repair process for the analysis of the impact of each fault in the process in order to solve a multi-fault rush-repair problem (MRRP) in power distribution networks. A direct matrix converter with space vector modulation (SVM) and transistor commutation was specified as a Petri net in [12], which allowed the reachability of particular states to be checked. The dynamic behavior of power systems protections was evaluated with Petri nets in [13]. Colored Petri nets were considered as a modeling formalism for a protective device from a single-phase-to-ground short circuit with an automatic change of current setting in electrical

networks of 6–10 kV in a recently published conference paper [14]. A timed Petri net model expressing generically the networked behavior of photovoltaic systems was proposed in [15]. Resource-oriented Petri nets [16] were in turn used for scheduling, e.g., for cluster tools in semiconductor manufacturing [17] or for crude oil operations in refineries (hybrid, colored-timed Petri nets being an extension of resource-oriented Petri nets) [18]. Other promising approaches to the control of power and energy systems used machine learning (ML) [19], deep learning (DL) [20], reinforcement learning (RL), [21] or fuzzy logic [22].

The interpreted Petri nets that we use follow the notions formally introduced in 2019 for the specification of cyber-physical systems [23] (although there are also some other definitions of "interpreted Petri nets" in the literature, e.g., [24,25]). They have proved so-far to be suitable in that domain. In contrast to traditional Petri nets, the interpreted ones additionally contain a description of the binary input and output signals of the system to communicate with the environment, with other systems, or their components. Unlike some other types of Petri nets that might also be useful, interpreted Petri nets do not involve inhibitor arcs, enabling arcs, or colored tokens, which enhances their simplicity of use. Input signals are assigned as guards to transitions, while output signals are assigned to places. What distinguishes interpreted Petri nets from ordinary Petri nets is that they are safe, which means that a place may contain only one token. An active place (that includes a token) indicates then directly the activity of the output signal assigned to it. Hence, it is also easy to generate some implementation code from them, e.g., targeted at AVR microcontrollers or FPGA devices [26]. An important issue then becomes determinism, a property of a model (not of a physical realization) meaning that it is not possible for the model to react in two or more ways to the same conditions [27]. The modeling methodology for a deterministic system specified by an interpreted Petri net is proposed in [28], distinguishing between strong and weak determinism.

There are several methods that allow verification of the control algorithm in power systems [29], including simulations, model checking (both symbolic and statistical), hardware in loop (HiL), and experiments. The specification of the control algorithm in a power and energy system is not only for its documentation but can also be used for validation before implementation. Formal specification by means of interpreted Petri nets offers some support by existing analysis and verification methods, which allows the checking of the basic properties at an early stage of development. Hence, possible errors related to the structure of a formal model may be resolved. Additionally, model checking can be performed to verify some behavioral properties. This aspect is, however, not explored further in this article.

This work continues our previous research on the use of interpreted Petri nets for the specification of cyber–physical systems (in particular their control parts) [23]. We now extend the application area and show that they can also be successfully used for autonomous components within power and energy systems. The presented example describes an autonomous system with electricity storage for the provision of a renewable energy source (RES) stabilization system service or one having highly variable loads. The analysis by the use of interpreted Petri nets takes into account measurement signals from the power system, diagnostic signals from the battery management system (BMS), and the relationships specified in the control algorithm for the power stabilization system service.

The main contributions of the paper can be summarized as follows:

(1) We show how interpreted Petri nets can be applied for autonomous components within power and energy systems;

(2) A novel modeling methodology for control algorithms in the power area with the use of interpreted nets is proposed that allows verification of basic properties at an early stage of system development;

(3) The presented idea is illustrated with a case study of an energy storage system;

(4) The possibilities of interpreted Petri nets in the energy domain are indicated, as are some implications for practical projects.

The remainder of the paper is structured as follows. Section 2 presents some background on interpreted Petri nets. Section 3 introduces a novel method for modeling power and energy system components with interpreted Petri nets. Section 4 illustrates the proposed approach with a case study. Section 5 presents experimental verification. Section 6 highlights the benefits and implications for practical projects. Finally, Section 7 summarizes and concludes the paper.

### 2. Background on Interpreted Petri Nets

For easier understanding and reading, some preliminaries are introduced.

**Definition 1.** *A Petri net [1] is a four-tuple PN = (P, T, F, $M_0$), where P is a finite set of places, T is a finite set of transitions, F ⊆ (P × T) ∪ (T × P) is a finite set of arcs and $M_0$ is an initial marking. A marking involves all places that contain a token. A transition is enabled in marking M, if each of its input places contains a token. A transition can be fired if it is enabled. Then, a token is removed from all its input places and added to all its output places. A marking is reachable from any other marking if it can be reached by a sequence of transition firings.*

**Definition 2.** *A Petri net is live [1] if it is always possible to fire any transition of the net by progressing through some further firing sequence.*

**Definition 3.** *A Petri net is safe [2] if there is no reachable marking such that any place contains more than one token.*

**Definition 4.** *A Petri net is reversible [2] if a return to the initial marking is always possible.*

Sample Petri nets with various properties are shown in Figure 1. The set of places and transitions is the same for all examples, each one consisting of three elements: $P$ = {$p1$, $p2$, $p3$} and $T$ = {$t1$, $t2$, $t3$}. The initial marking is also the same, $M_0$ = {$p1$}. The only difference is in set $F$, which includes the relations between places and transitions. This aspect influences the basic properties.

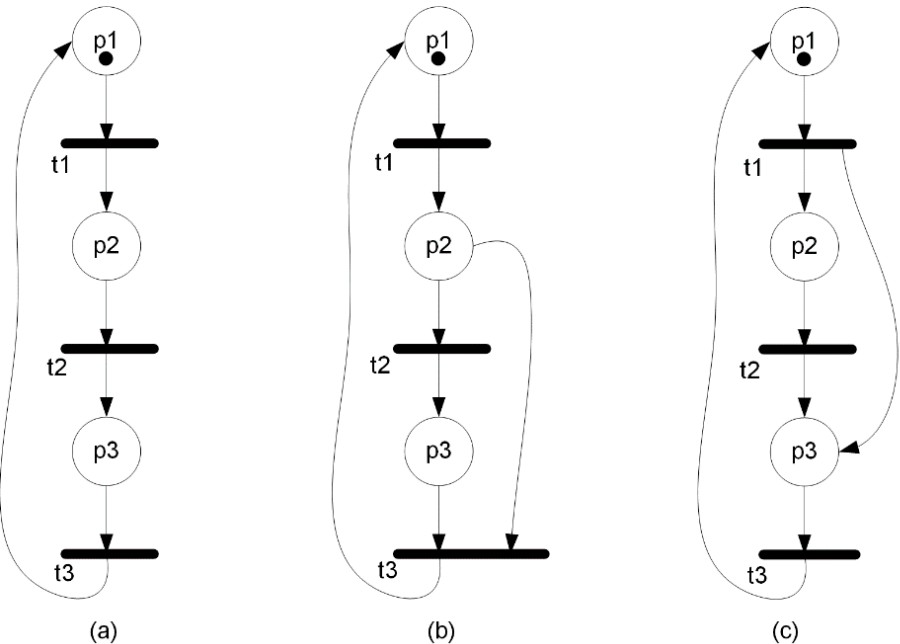

**Figure 1.** Petri nets and their structural properties: (**a**) live, safe, and reversible; (**b**) safe but not live; (**c**) live but not safe.

**Definition 5.** *An interpreted Petri net [23] is a six-tuple IN = (P, T, F, $M_0$, X, Y), where the first four elements describe a Petri net that is live and safe; X is a finite set of logic input signals and Y is a finite set of logic output signals. A transition in an interpreted net can be fired if it is enabled and all the conditions of its input signals (assigned to it) are fulfilled.*

To illustrate the above definitions, a sample Petri net is presented in Figure 2a, while an interpreted Petri net (with the same structure but extended with input and output signals) is shown in Figure 2b. Both nets contain four places and four transitions connected with each other. However, the ordinary Petri net (Figure 2a) does not include any additional information, while the interpreted Petri net (Figure 2b) takes into account input and output signals for communication with the environment. In the example, the interpreted net models the functionality of a two-bit counter, counting from zero (binary 00) to three (binary 11), showing the current number with light-emitting diode (LED) lights. One input signal (*go*) is used to proceed with counting (if it equals 0, then the counter freezes). Two output signals are used to control the LEDs (*y*1 and *y*0). Initially, the counter is set to zero (place *p*1 contains a token) and the LEDs are turned off. With active signal *go*, the counting proceeds, transition *t*1 fires (as its input place contains a token and its guard is fulfilled) and the appropriate LED is turned on.

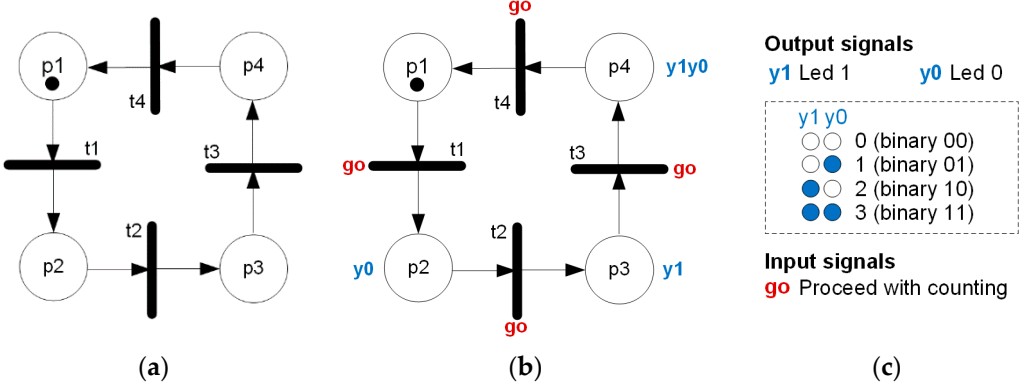

|                  |                  |                  |
| :--------------: | :--------------: | :--------------: |
|       (**a**)    |      (**b**)     |      (**c**)     |

**Figure 2.** A sample Petri net (**a**), an interpreted Petri net (**b**), and signals description (**c**).

**Definition 6.** *An interpreted Petri net is weakly deterministic [28] if for each reachable state (marking) and for any fixed input value, the net comes into a stable marking and at the same time there is no stable marking into which the net can come with the same input values.*

**Definition 7.** *An interpreted Petri net is strongly deterministic [28] if it is weakly deterministic and for each reachable marking and any fixed input values there is only one next marking possible.*

### 3. Modeling Autonomous Components within Power and Energy Systems with Interpreted Petri Nets

The proposed methodology for modeling the autonomous components within power and energy systems is briefly illustrated in Figure 3 and includes six steps. The schema graphically shows the proposed flow and is strongly connected with Algorithm 1.

First, output signals corresponding to system states should be specified. They are used to indicate the current system states (e.g., operation mode or progress in algorithm realization). Then, input signals corresponding to conditions should be defined. They are used as triggers between system states and indirectly influence the property of determinism (q.v. Definitions 6 and 7), which will be shown subsequently. Next, the structure of an interpreted Petri net can be created. It is used as a basis specification, which is supplemented by the defined input and output signals. It influences directly the properties of liveness, safety, and reversibility (q.v. Definitions 2–4). Output signals are assigned to places and used for activating appropriate signals related to the current status of algorithm realization.

In other words, the particular output signal assigned to a place is active as long as that place contains a token. Input signals are assigned to transitions and used as conditions for their particular firing, and they indirectly influence the property of determinism (q.v. Definitions 6 and 7). If the guard of a transition is fulfilled and its input place(s) contain(s) a token, the transition may be fired. Finally, we obtain the interpreted Petri net, which can be analyzed and formally verified, focusing especially on such properties as liveness, safety, reversibility, and determinism, which can be treated as performance indicators.

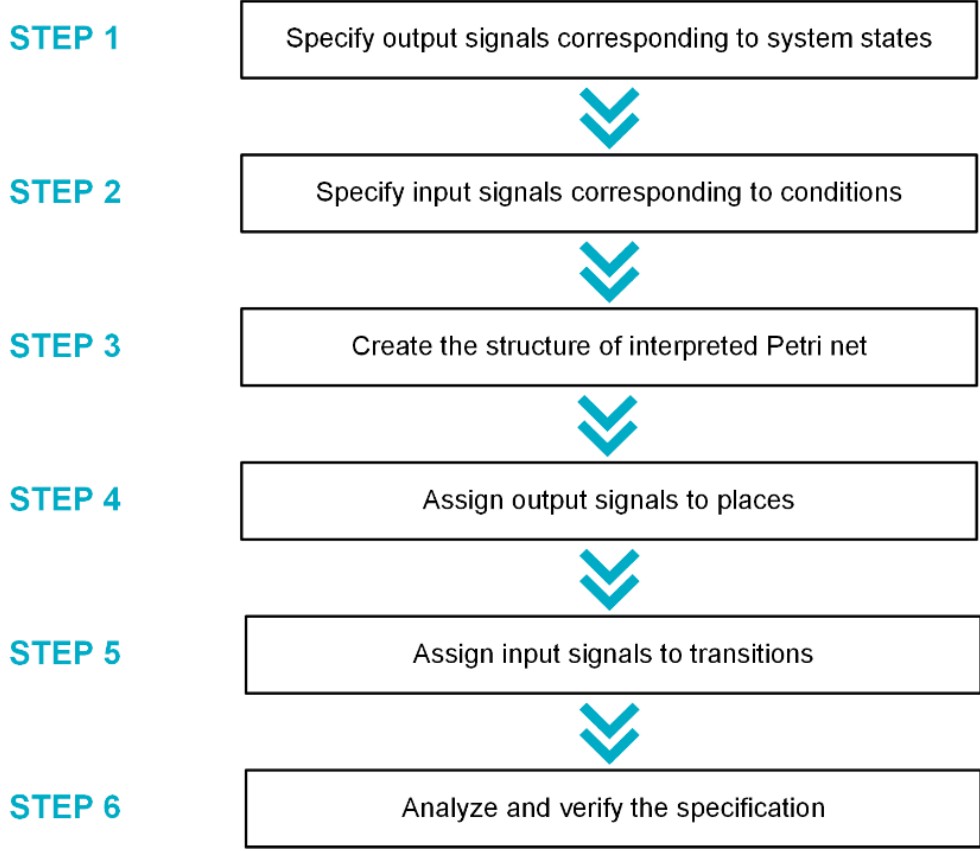

**Figure 3.** The schema of the proposed modeling methodology.

Formally, construction of an interpreted Petri net for autonomous components within power systems, focusing in particular on the control aspects, can be realized following Algorithm 1. The starting point is an informal specification, while the result is a properly formed interpreted Petri net. The algorithm realizes steps 1–5 of the proposed modeling methodology, formally defining step 1 (line 1), step 2 (lines 2–5), step 3 (lines 6–8), step 4 (lines 9–11), and finally, step 5 (lines 12–16). As a result, an interpreted Petri net is returned (line 17) that can be analyzed and verified (step 6).

The structure of the constructed interpreted Petri net (the result of step 3) directly influences the properties of liveness, safety, and reversibility (q.v. Definitions 2, 3 and 5). In order to achieve a live net, no deadlock can occur. In order to achieve a safe net, all places may contain at most one token. In order to achieve a reversible net, the initial marking has to be reachable from any other marking.

The property of determinism is directly influenced by the structure and input signals which are assigned to transitions. If two transitions with a common input place have non-exclusive conditions, it is not known which of them will be realized if both conditions are true (illustrated in Figure 4a: if $x1 = 1$ and $x2 = 1$, either transition $t1$ or $t2$ will fire).

| **Algorithm 1:** Construction of an interpreted Petri net |
| --- |
| **Input:** an informal specification of an autonomous component within a power system <br> **Output:** an interpreted Petri net $IN = (P, T, F, M_0, X, Y)$ |

| | |
| --- | --- |
| 1: | Identify the steps of control algorithm realization and specify set $Y$ of output signals to show the current state (progress); |
| 2: | Identify the conditions for control algorithm realization; |
| 3: | Create an empty set, $E$, of elementary expressions; |
| 4: | Split the conditions into elementary expressions, $e$, and add them to the set $E$; If set $E$ already contains expression $e$, then check the next condition; |
| 5: | Specify set $X$ of input signals by assigning each $e \in E$ to a separate signal name; |
| 6: | Create the structure of the Petri net $PN$ by adding places $p \in P$, transitions $t \in T$ and arcs $f \in F$, connecting them with each other to reflect the control algorithm; |
| 7: | Define the intial marking, $M_0$, by specifying where the algorithm starts; |
| 8: | A Petri net $PN = (P, T, F, M_0)$ is obtained; |
| 9: | **for each** $p \in P$ **do;** |
| 10: | Assign to $p$ an output signal (or more output signals) $y \in Y$, if necessary; |
| 11: | **end for;** |
| 12: | **for each** $t \in T$ **do;** |
| 13: | Check whether its firing should be guarded; |
| 14: | If not, consider the next transition, $t$ |
| 15: | Otherwise, assign the appropriate expression using defined input signals $x \in X$ and elementary Boolean operators (and/or/negation); |
| 16: | **end for;** |
| 17: | Return an interpreted Petri net $IN = (P, T, F, M_0, X, Y)$. |

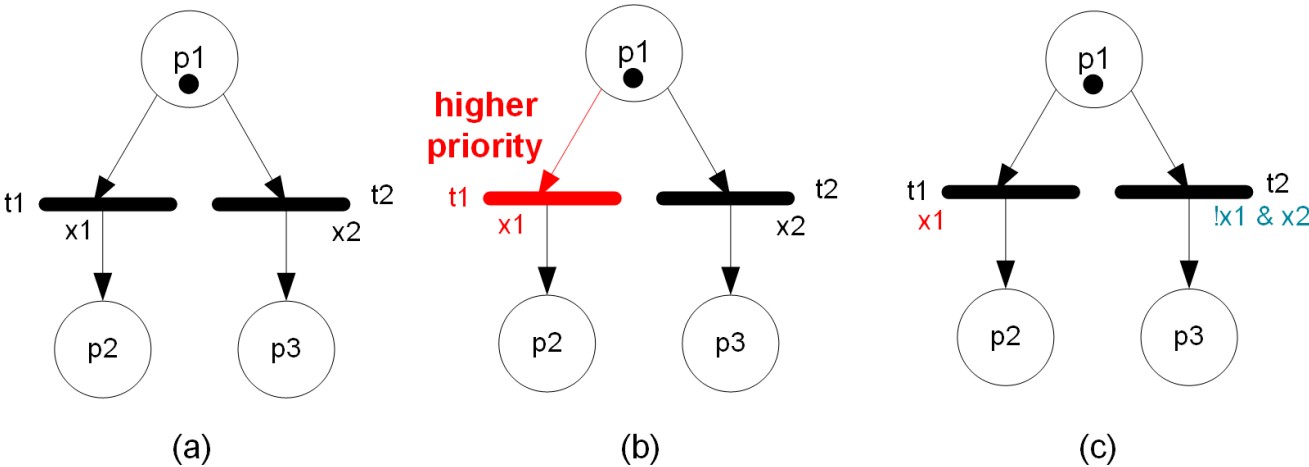

**Figure 4.** Input signals influencing the property of determinism: (**a**) non deterministic interpreted Petri net; (**b**) showing the priority; (**c**) deterministic (more precisely—strongly deterministic) interpreted Petri net.

The appropriate assignment of conditions, also taking into account conflicting transitions results in the determinism. Algorithm 2 formally describes how to resolve the conflict among transitions.

Following the algorithm, after identifying the transition with higher priority (line 1, Figure 4b), the appropriate logical expression is built (line 2) and assigned to the transition with the lower priority (Figure 4c). Finally, we can achieve strong determinism

| **Algorithm 2:** Resolving conflicts among transitions |
|---|
| **Input:** two transitions in conflict, with distinct guards (logically) <br> **Output:** two transitions without conflict |
| 1:　Identify the transition with a higher prority. Let it be *t_high*. Let the other one be *t_low*; |
| 2:　Form a conjuction of the guard from transition *t_low* with the negation of the guard from transition *t_high*; |
| 3:　Change the guard of transition *t_low* to be the logical expression from step 2; |
| 4:　Do not change the guard of transition *t_high*; |
| 5:　Return transitions *t_low* and *t_high*. |

## 4. Case Study

As a case study, an energy storage system is used to provide a system service related to the stabilization of power resulting from fluctuations in the power generated by RES or loads with large and sudden power changes [30,31]. Firstly, it is informally specified (Section 4.1). Then, it is modeled as an interpreted Petri net (Section 4.2). Afterwards, the model is analyzed to check its basic properties, and necessary modifications are introduced so that all important properties are satisfied (Section 4.3).

### 4.1. Informal Specification

The power stabilization algorithm is initiated when the permissible (limit) power fluctuations at the connection point are exceeded $\Delta P_{RES/LOAD} > P_{Lim}$ [30]. The change in power in the grid may be caused by the variability of energy production from RES or the variability of the load. RES are characterized by the variability of the produced energy resulting from changing weather conditions, e.g., the solar or wind power plant. On the other hand, loads that significantly change their value in a short time are, for example, the starting of high-power drives, induction welders, or welding machines. The operation of the algorithm is shown in Figure 5. In the case of the implementation of this algorithm, it is necessary to maintain the level of charge (SoC) of the energy storage system allowing both charging and discharging. Before starting the process of stabilizing power fluctuations, the charge level SoC of the energy storage system is checked. Taking into account the operating conditions of the energy storage system (minimum and maximum SoC level), it was assumed that the required SoC level for the implementation of the power stabilization algorithm should be 60–65% $SoC_{max}$ ($SoC_{max}0.6 < SoC < 0.65\ SoC_{max}$). Such a SoC level enables both the transfer of energy to the power grid (discharge of the energy storage) and the consumption of energy by a storage system (charging an energy storage system). If the SoC level is higher than $0.65\ SoC_{max}$, the energy storage system is discharged with the assumed power $P_{ES} = P_{DCH}$ (discharge power) until the required level of SoC. When the SoC level is lower than $0.6\ SoC_{max}$, the energy storage system is charged with $P_{ES} = P_{CH}$ (charging power). When the required SoC level is achieved, the power stabilization procedure can be started.

The power stabilization algorithm is a follow-up control system in which the set value (the power to be maintained at the PCC connection point) changes in a manner unknown in advance. The purpose of the control system is to follow the changes of the set value. The algorithm assumes maintaining a constant power ($P_{ref}$) at the PCC for 1-minute periods (time $t_1$). The value of the power that should be maintained $P_{ref}$ in the next minute is determined on the basis of the average RES/LOAD power that was recorded in the previous period (1).

At time $t_1$, the difference between the line power, measured at the PCC point ($P_{RES/LOAD}$), and the power $P_{ref}$, is determined. The difference between $P_{ref}$ and $P_{RES/LOAD}$ is covered from the energy storage system. After the time $t_1$ has elapsed, the power $P_{ref}$ is determined again for the next period.

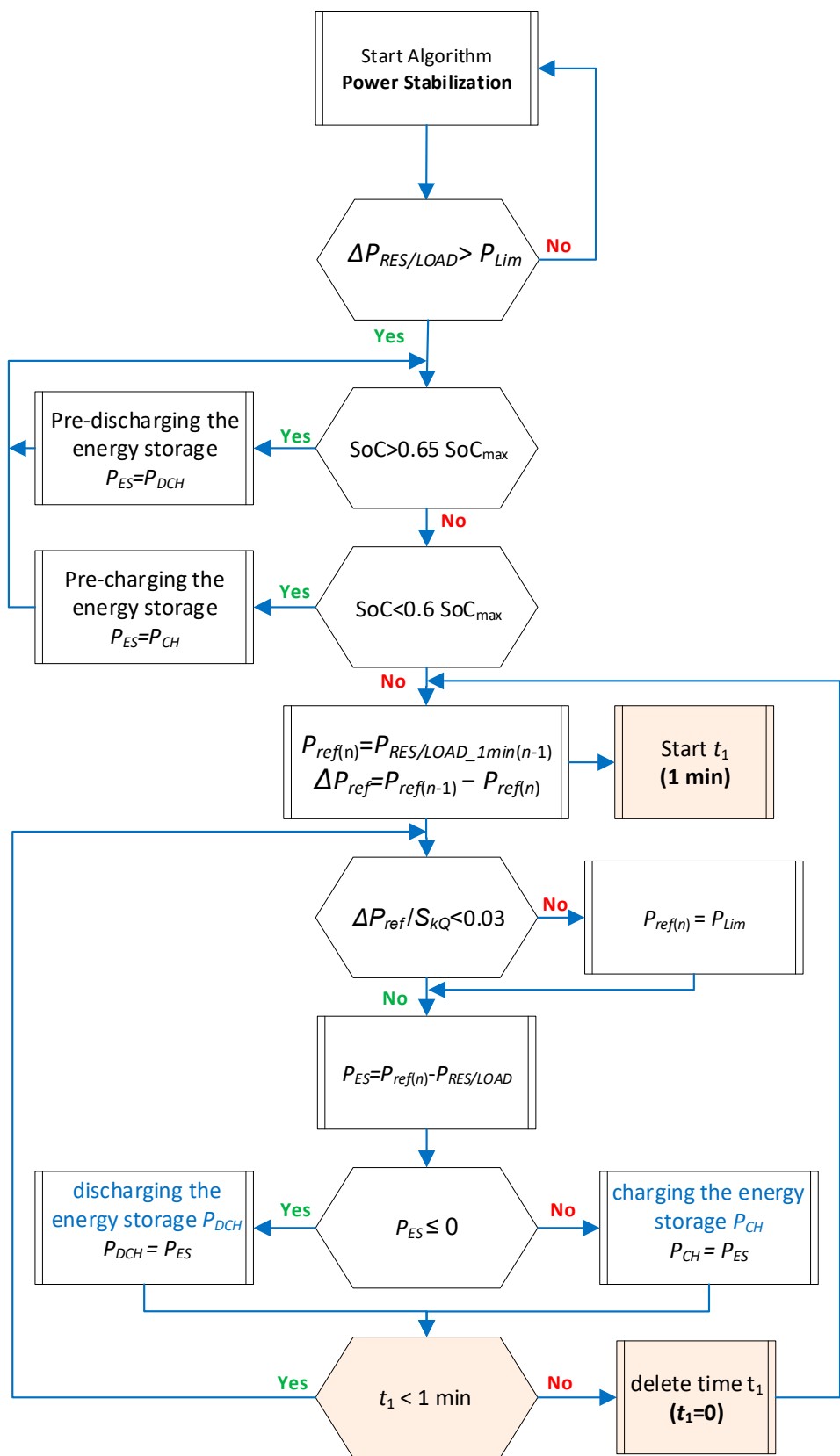

**Figure 5.** General algorithm for power stabilization of RES and loads.

$$P_{ref(n)} = \overline{P}_{RES/LOAD\_1min(n-1)} \tag{1}$$

The algorithm also takes into account the maximum allowable power change $\Delta P_{ref}$ (2) so as not to adversely affect the flicker $P_{st}$ coefficient (3). This means that a step change in voltage occurring within one minute must not have an effect on the power grid in the form of a relative voltage change of more than 3% ($\Delta U / U_N < 3\%$).

$$\Delta P_{ref} = P_{ref(n-1)} - P_{ref(n)} \tag{2}$$

$$\frac{\Delta P_{ref}}{S_{kQ}} < 0.03 \tag{3}$$

where $S_{kQ}$ is a short-circuit power at the PCC. If $\Delta P_{ref} \geq P_{Lim}$ then the algorithm will limit the charging/discharging power of the energy storage system to $P_{ES} = P_{Lim}$. If $\Delta P_{ref} < P_{Lim}$ then the charging/discharging power of the energy storage system will result from the power difference $P_{RES/LOAD}$ measured at the PCC point and the power $P_{ref(n)}$ ($P_{ES} = P_{ref(n)} - P_{RES/LOAD}$).

Depending on the type of energy storage system used, the control algorithm may include operating procedures, if required (the maximum charging and discharging currents, which are almost always different for most electricity storage technologies; cell temperature and air conditioning in summer, winter heating, etc.). These are not considered further in this article.

*4.2. Formal Specification*

In order to formally specify a detailed description of the algorithm's operation, an interpreted Petri net model was created (Figure 6), which takes into account additional dependencies related to the physical constraints of the system. This model defines the transition conditions between the respective operating states of the system. First, the storage system must provide the initial conditions for the provision of the power stabilization service—$0.6\,\mathrm{SoC_{MAX}} \leq \mathrm{SoC} \leq 0.65\,\mathrm{SoC_{MAX}}$. After reaching the initial conditions, then the power stabilization algorithm is implemented, according to the relationships (1)–(3) and the diagram in Figure 5. Then, the operating points associated with discharging or charging the energy storage system, with a power $P_{ES}$ lower than $P_{Lim}$ or a power $P_{ES}$ greater than $P_{Lim}$, are defined. It is always possible to disable the provision of the power stabilization service. In the occurrence of full charge ($\mathrm{SoC} = \mathrm{SoC_{MAX}}$) or full discharge ($\mathrm{SoC} = \mathrm{SoC_{MIN}}$), for the algorithm to continue to work, the discharge process $P_{ES} < 0$ or the charging process $P_{ES} > 0$ must follow. If there is a state of full charge or discharge of the energy reservoir, the algorithm will not be able to fully compensate for power fluctuations during this time. The transition of the energy storage system into the operating state with power limitation to $P_{Lim}$ occurs when the system is installed in a network with low short-circuit power, Equation (3). Then, the charge power or discharge power of the energy storage system could affect excessive high voltage fluctuations in the network. The transition to this operating state of the signal is associated with large power changes in the system with low short-circuit power (high short-circuit loop impedance).

Following the proposed modeling methodology, the general steps of control algorithm realization have been identified and appropriate output signals have been defined that correspond to the system state (line 1 of Algorithm 1). They are used to indicate the current status of the system. The elements of set $Y$ are listed in Table 1. Next, the conditions and input signals for the control algorithm have been defined (lines 2–5 of Algorithm 1) that directly influence the current system state. The input signals are *true* if the corresponding conditions are fulfilled. Otherwise, they are *false*. The elements of set $X$ are listed in Table 2.

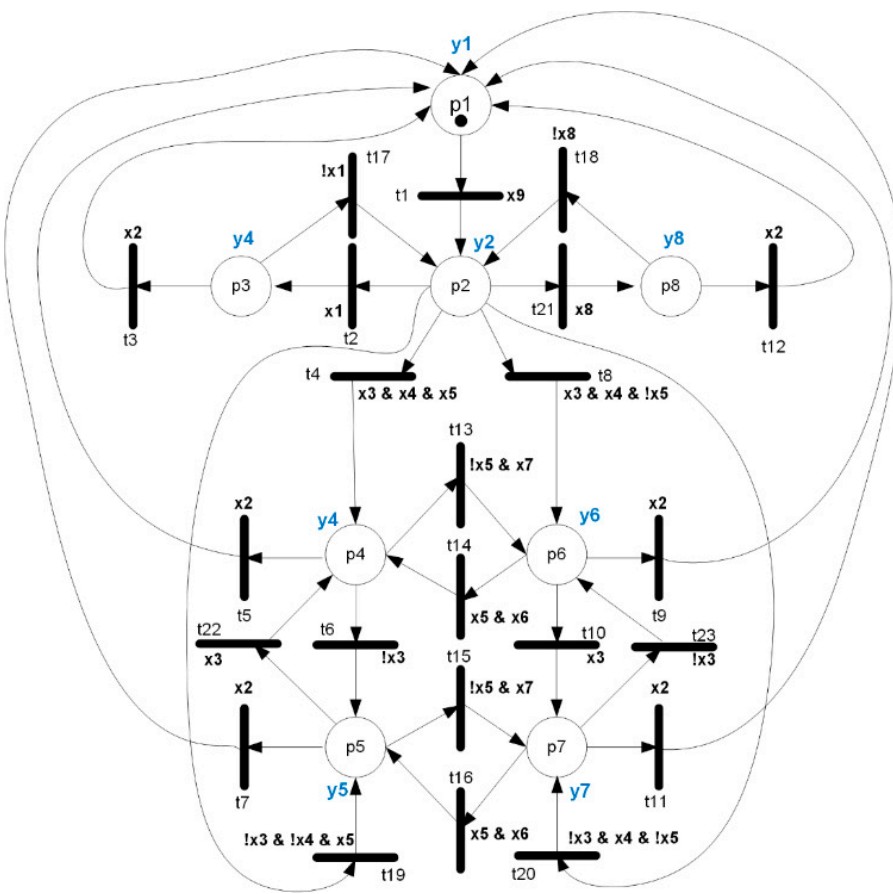

**Figure 6.** An interpreted Petri net describing directly the algorithm of power stabilization of RES and loads.

**Table 1.** Binary output signals (set *Y*).

| Signal | Description |
|---|---|
| y1 | Idle state |
| y2 | Initialization of initial conditions of the algorithm |
| y3 | Battery pre-charging |
| y4 | Battery charging with $|P_{ESmax}| \leq |P_{Lim}|$ |
| y5 | Battery charging with $|P_{ESmax}| = |P_{Lim}|$ |
| y6 | Battery discharging with $|P_{ESmax}| \leq |P_{Lim}|$ |
| y7 | Battery discharging with $|P_{ESmax}| = |P_{Lim}|$ |
| y8 | Battery pre-discharging |

**Table 2.** Binary input signals (set *X*).

| Signal | Description (Condition to Be True) |
|---|---|
| x1 | $SoC \leq 0.6\ SoC_{MAX}$ |
| x2 | Request to quit the algorithm |
| x3 | $\Delta P_{ref}/S_{kA} \leq 0.03$ |
| x4 | $\Delta P_{RES/LOAD} \leq P_{Lim}$ |
| x5 | $P_{ES} > 0$ |
| x6 | $SoC = SoC_{MIN}$ |
| x7 | $SoC = SoC_{MAX}$ |
| x8 | $SoC \geq 0.65\ SoC_{MAX}$ |
| x9 | Request to start the algorithm |

Afterwards, the structure of the interpreted Petri net is created so that it properly describes the functionality (line 6 of Algorithm 1). The initial marking involves place *p1* and corresponds to the idle state (line 7 of Algorithm 1). Places are connected with each other via transitions, some of which are mutually sequential to each other. Next, output signals are assigned to appropriate places as system states (lines 9–11 of Algorithm 1), e.g., signal *y*1 (indication of idle state) is assigned to place *p1,* which means that it is active as long as place *p*1 maintains a token. In turn, input signals are assigned to transitions as their guards (lines 12–16 of Algorithm 1), e.g., signal *x9* (request to start the algorithm) is assigned to transition *t*1. A transition is fired (realized) if its incoming place has a token and the condition assigned to it is true, e.g., transition *t*1 is fired if place *p*1 contains a token and input signal *x*9 is true, then a token is removed from place *p*1 and added to place *p*2. Otherwise, the net does not change its marking (i.e., it does not evolve). The complete interpreted Petri net describing the overall supervisory energy management system of the algorithm for the stabilization power of RES/LOAD is shown in Figure 6 (resulting from line 17 of Algorithm 1).

*4.3. Analysis of Specifications and Required Modifications*

The formal specifications, in the form of an interpreted Petri net, may then be analyzed and verified to check its basic properties using analytical or verification methods. Some properties may be checked by application of available Petri net tools. The most important properties include:

(a)   liveness, i.e., whether or not the model gets stuck in a deadlock state;
(b)   safety, i.e., whether all places contain at most one token;
(c)   reversibility, i.e., whether it is possible to return to the initial state;
(d)   determinism, i.e., whether the net is deterministic (weakly or strongly).

Thus, the analysis of the interpreted Petri net from Figure 6 reveals that it is live, safe, and reversible. The initial marking is reachable from any other marking in the net and the algorithm does not get stuck.

However, the Petri net is not deterministic. For example, when place *p3* is active, two transitions may be realized—either *t3* with guard "*x2*" or transition *t17* with guard "!*x1*" (in the case when *x*1 is false and *x*2 is true). If only one of these conditions is fulfilled, the net behaves in a deterministic way. Otherwise—if two of them are fulfilled ($SoC \leq 0.6$ $SoC_{MAX}$ and a request to quit the algorithm is sent)—we cannot be sure which of them will be realized. This is obviously not a desired property, so we have to adjust the guards of the transitions so that they are mutually exclusive. The highest priority of the system's operation is always the signal to disable the provision of the power stabilization service (in our case it is signal *x*2).To do so, the priorities must be given to transitions (according to the proposed Algorithm 2), which result in more complex guards of transitions. For example, conflicting transitions *t3* and *t17* are then resolved by changing the guard of transition *t17* to "!*x*1 & !*x*2" and giving a higher priority for a request to quit the algorithm. Moreover, the charging and discharging of various parameters is dependent on the value of $\Delta P_{ref}/S_{kA}$, and therefore the guards of transitions outgoing from those places are also extended to be strongly deterministic and to show the priorities. It should be noted that not all transitions are really in conflict, although when looking at the structure of the Petri net, it seems that this may be the case. Transition *t2* and *t21* are not in conflict, despite the fact that they are assigned just simple input signals. Here, it is not possible that input signals *x*1 (guard of *t*2) and input signal *x8* (guard of *t21*) can be active at the same time, because either $SOC \leq 0.6$ $SOC_{MAX}$ or $SOC \geq 0.65$ $SOC_{MAX}$ (see also Table 2). The revised deterministic interpreted Petri net is presented in Figure 7.

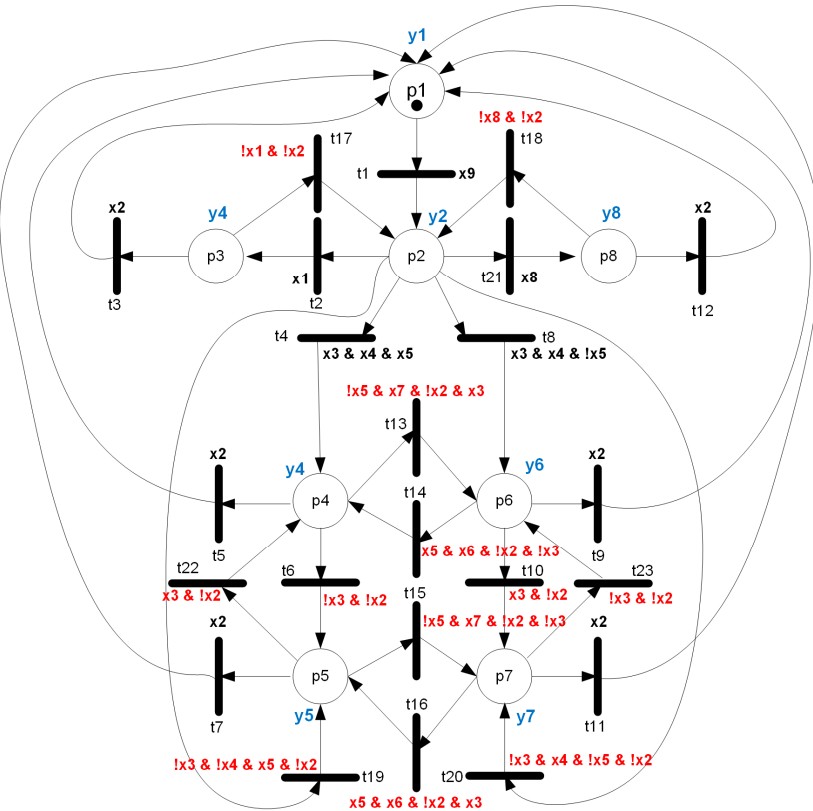

**Figure 7.** A revised strongly deterministic interpreted Petri net describing the algorithm of power stabilization of RES and loads.

## 5. Experimental Verification

Experimental research was carried out on a system with an energy storage unit with batteries made from LTO (lithium–titanium-oxide) technology (Figure 8) [31]. The LTO-type energy storage system design is based on the rechargeable battery SCiB 23Ah cell manufactured by Toshiba. According to the catalog note, the estimated cell life is 20,000 cycles with a 70% DoD discharge rate. The power of the tested electricity storage was 100 kW and its capacity was 35 kWh. The continuous charge/discharge current at 25 °C is 200 A. The electronic power bidirectional DC/AC converter connecting the energy storage system with the power grid is made of two 50 kW inverters working in parallel [32]. The inverters are built using SiC technology, with galvanic isolation and high-frequency transformers (25 kHz).

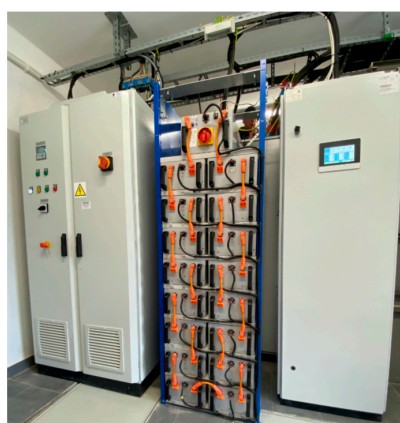

**Figure 8.** A prototype of a 100 kW energy storage system using LTO technology.

The prototype of the energy storage system was installed at a transformer station and connected via fuses and main switch directly to the terminals of the MV/LV transformer with a rated power of 630 kVA. The short-circuit power $S_{kQ}$ at the point of connection of the energy storage system is much greater than the power that can be generated by the storage, so in the tested system the criterion described by Equation (3) is always satisfied and there is no limitation of storage generation to $P_{Lim}$.

The study of the load power stabilization algorithm was carried out in two tests using the Fluke 437-II Network Parameter Analyzer. In the first test, one large receiver with a power of 18 kW was switched on, which changed its power from zero to nominal value within a minute. Additionally, other loads with variable power change characteristics were accidentally switched on, due mainly to the start-up of asynchronous motors or their operation with variable load. The general profile of the network load changes also includes the loads that normally operate in the network. In the second test, one large receiver with a power of 18 kW was turned on, which changed its power from zero to nominal value within a minute. Moreover, the loads that normally operated in the network were also included in the overall profile of the network load changes.

The test results for the first test are shown in the Figure 9a. The compensation algorithm was activated at the time $t_0$. Before starting the algorithm, the energy storage system was set to the appropriate state of charge $0.6 \, \text{SoC}_{MAX} \leq \text{SoC} \leq 0.65 \, \text{SoC}_{MAX}$. As a result of the algorithm of the load power compensation using an energy storage system unit, the power consumed from the grid did not exceed 40 kW, with low-frequency fluctuations not exceeding 15 kW. While the load power was lower than the average reference power $P_{ref\,(n-1)}$, the energy storage system was charged from the grid ($P_{ES}$ with the "+" sign). In this way, the energy storage system kept the power within the calculated average power $P_{ref(n-1)}$. In time intervals when the load power was higher than the $P_{ref(n-1)}$, the energy storage system returned energy to the grid ($P_{ES}$ with the "−" sign), reducing the power from the grid to the determined average value $P_{ref(n-1)}$. The presented research results show a lack of compensation for fast-changing fluctuations in the load power. This is due to the measuring system used in the prototype cooperating with the control system. The measurement data acquisition time of 250 ms is too long to effectively stabilize the fast-changing load in the power grid.

The test results for the second test, with a slowly changing load, are presented in Figure 9b. It is also noteworthy that there was no full compensation when the load changed rapidly. This is due to the relatively long acquisition time of data measurement of 250 ms. In order to eliminate this unfavorable property, it should be ensured that the acquisition of measurement data from the power grid and the implemented compensation algorithm are carried out in the system in which the control of the electronic power converter is carried out. These operations must be performed with correspondingly shorter sampling times. In the tested prototype, these two control algorithms are separated. The communication time between these systems is 250 ms, which is too long for compensation tasks for a supply voltage period of 20 ms.

As the prototype of the energy storage system was built as an autonomous, maintenance-free device, it was not possible to measure signals regarding the SoC level and the set average power values. Nevertheless, the implemented algorithm was verified in the testing phase using the method proposed in the article. Unfortunately, not all output parameters of the tested system were made available for measurement.

The experimental test of the system was carried out only for load changes. Changes in RES generation were not possible in our laboratory. Had that been the case, the difference in the operation of the algorithm would be that the energy storage system with too much generation from RES would charge, and in the case of reduced generation from RES, it would discharge.

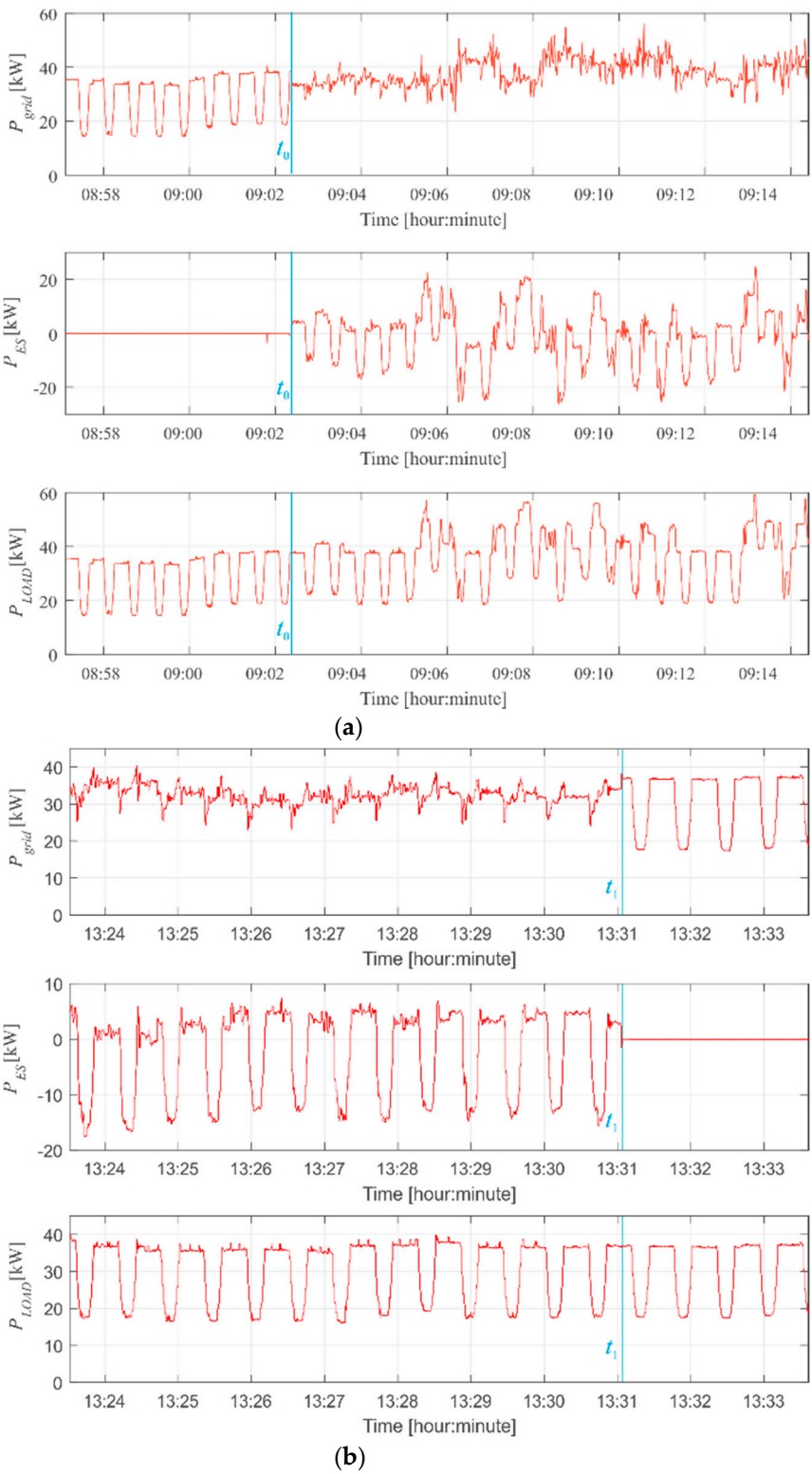

**Figure 9.** Time waveforms of power consumed from the grid $P_{grid}$, power of the energy storage system $P_{ES}$ and load power $P_{LOAD}$ for the operation of the power stabilization algorithm of unstable loads with LTO energy storage, (**a**) at a fast changing load (visualization of the moment of switching-on the algorithm) and (**b**) at a slowly changing load (visualization of the moment of switching-off the algorithm).

## 6. Discussion

### 6.1. Possibilities of Interpreted Petri Nets

Interpreted Petri nets take the benefits of ordinary Petri nets as a mathematical formalism with all available analysis and verification methods. Additionally, they are equipped with input and output signals that allow communication with the environment. When considering the control algorithms of autonomous components within power and energy systems, such a formal specification can be easily verified before implementation. This allows us to detect such problems as deadlocks or indeterminism, which was shown in the case study, and to adjust the structure so that the important properties hold.

Moreover, based on a formal specification in the form of a previously verified interpreted Petri net, we can automatically generate a program in a hardware description language (e.g., VHDL or Verilog). Therefore, the implementation of a prototype can be obtained in a short time.

To summarize, the benefits for applying interpreted Petri nets are as follows. Firstly, an autonomous component in an electrical power system is formally specified. Secondly, the prepared model can be formally verified in order to check some basic properties, e.g., determinism or liveness. Behavioral properties can also be verified using a model checking technique. Thirdly, code in a hardware description language can be automatically generated. These advantages of interpreted Petri nets also make them suitable for the modeling of autonomous components within power and energy systems.

### 6.2. Implications for Practical Projects

An advantage of the proposed modeling technique is the application of formal methods [33] which take into consideration the specification as well as the analysis and verification. As a result, the development time and the cost can be reduced, since errors may be detected at an early stage. Having formal specification (in the form of interpreted Petri net), symbolic model checking can be used to verify the mutual relationship between input and output signals.

An inconvenience, in turn, is that knowledge of interpreted Petri net formalism is needed, as is knowledge of the possible analysis and verification methods that can be applied. However, this barrier can be easily overcome with some interdisciplinary research involving both power and energy systems specialists and automatic control engineers. Moreover, an interpreted Petri net model is always some kind of abstraction. It means that it presents only a view of the system from some chosen perspective, and certain detailed information may intentionally be omitted at this stage of system development.

Usually, when designing control algorithms in autonomous components within energy systems, no formal methods are used (or at least very rarely [29]), and the proposed solutions are just simulated to check for proper behavior. However, neither simulations or tests can confirm that the designed system will behave correctly in all situations, and these methods take into account only some pre-defined scenarios. The application of formal methods (including model checking) allows for more confidence regarding functionality. The challenge here is therefore to popularize formal methods in the electric power system domain in order to benefit from the diverse possibilities of interpreted Petri nets (as a formal model) that can be easily formally verified using, e.g., a model checking technique.

## 7. Conclusions

In this article, we have shown how interpreted Petri nets can be applied to power and energy system specification, considering, in particular, the control parts of autonomous components. The main benefits of using this kind of formalism have been highlighted, as well as the possible implications for practitioners. It should be noted that interpreted Petri nets, originally dedicated to the specification of the control parts in cyber–physical systems, may also be successfully used in other domains, such as energy systems. They are well suited for modeling sequential and concurrent operations as well as for sharing

of resources. Using Petri nets as a modeling technique, one can then benefit from formal analysis and verification methods.

The limitations of the proposed modeling technique include the need for cooperation with interdisciplinary engineers so that the specific aspects of the control algorithms in the power and energy systems can be modeled in a way that is closer to the automatic and control domain. Plans for the future therefore include closer interdisciplinary cooperation to introduce more advancements into the area of power and energy systems.

**Author Contributions:** Conceptualization, I.G. and P.S.; methodology, I.G.; research, I.G.; writing—original draft preparation, I.G. and P.S.; writing—review and editing, I.G. and P.S.; visualization, I.G. and P.S., experimental verification, P.S. All authors have read and agreed to the published version of the manuscript.

**Funding:** The work is supported by The National Science Centre, Poland (grant number 2019/35/B/ST6/01683).

**Institutional Review Board Statement:** Not applicable.

**Informed Consent Statement:** Not applicable.

**Data Availability Statement:** This study did not report any data.

**Conflicts of Interest:** The authors declare no conflict of interest.

## Nomenclature

| | |
|---|---|
| *DoD* | depth of discharge, the level of discharge of a battery relative to its capacity |
| *PCC* | point of common coupling |
| $P_{CH}$ | set power to charge the energy storage |
| $P_{DCH}$ | set power to discharge the energy storage |
| $P_{ES}$ | power consumed/delivered from/to the energy storage system |
| $P_{Lim}$ | acceptable limit of power change in the system not causing deterioration of the electric power quality parameters |
| $P_{ref}$ | power set point at PCC |
| $\Delta P_{RES/LOAD}$ | change in power resulting from the variability of RES (renewable energy sources) generation or variability of the load power |
| $S_{kQ}$ | short circuit power at the PCC |
| *SoC* | the level of charge of a battery relative to its capacity |
| $P_{RES/LOAD\_1\,min}$ | average value of RES power/loads within one minute |
| $\Delta U$ | voltage change in the power grid |
| $U_N$ | nominal value of the voltage in the power grid |

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
