# Peer review of "Interpreted Petri Nets Applied to Autonomous Components within Electric Power Systems"

_applsci, doi:10.3390/app12094772_

Round 1

Reviewer 1 Report

The paper describes an application of interpreted Petri nets for modelling and verification of autonomous components within power energy systems. The article does not present a new modelling method based on using Petri nets. This is rather a case-based proof that this class of Petri nets can be successfully applied for developing submodules of power energy systems. The approach is well described and the provided results seem to confirm the usefulness of the considered approach. 

In general, the paper is well written and the arguments can be easily followed. It can be accepted for the journal, but a few minor improvements should be made:
1. Definition 2 is unclear because of the sentence "it is possible to fire any transition by a sequence of firings of other transitions."
2. Definition 3: replace "the place" with "any place".
3. Figure 1b: The net is not live. The t1 transition can be fired only once. 
4. I don't understand what the purpose of including Figure 3 is. All this information is in Algorithm 1.

Author Response

We would like to thank the reviewer for the valuable comments. We have carefully revised and improved the quality of the paper according to the given remarks. The changes in the manuscript are highlighted in blue. Below, we state point by point our responses to the comments, as well as the actions taken in the paper.

Comment #1: The paper describes an application of interpreted Petri nets for modelling and verification of autonomous components within power energy systems. The article does not present a new modelling method based on using Petri nets. This is rather a case-based proof that this class of Petri nets can be successfully applied for developing submodules of power energy systems. The approach is well described and the provided results seem to confirm the usefulness of the considered approach.
In general, the paper is well written and the arguments can be easily followed. It can be accepted for the journal, but a few minor improvements should be made.
Response: Thank you for your positive feedback.
Action: None

Comment #2: Definition 2 is unclear because of the sentence "it is possible to fire any transition by a sequence of firings of other transitions.”
Response: Thank you for pointing out this issue. We have exchanged the definition based on [2] with the definition from [1] as follows:
Definition 2. A Petri net is live [1], if it is always possible to fire any transition of the net by progressing through some further firing sequence.
Action: The changes in the manuscript involve lines 139-140.

Comment #3: Definition 3: replace "the place" with "any place".
Response: Thank you for the comment. We have changed the phrase.
Action: The changes in the manuscript involve line 142.

Comment #4: Figure 1b: The net is not live. The t1 transition can be fired only once. 
Response: Thank you for pointing out this issue. We have deleted the net from Fig. 1b.
Action: Figure 1 has been changed.

Comment #5: I don't understand what the purpose of including Figure 3 is. All this information is in Algorithm 1.
Response: Thank you for the comment. Indeed, Algorithm 1 and Figure 3 are strongly connected with each other. The purpose of including Figure 3 was to graphically and schematically present the main idea of the algorithm. In order to explain the motivation of introducing Figure 3 into the manuscript, we have added the following statement:
The schema graphically shows the proposed flow and is strongly connected with Algorithm 1.
Action: The changes in the manuscript involve lines 190-191.

Reviewer 2 Report

A more complete literature coverage should include also use of Timed Petri nets. (for example, Simon, D. F., Teixeira, M., & da Costa, J. P. (2022). Availability estimation in photovoltaic generation systems using Timed Petri Net simulation models. International Journal of Electrical Power & Energy Systems137, 106897.)

A more complete literature coverage should include also use of Fuzzy logic (for example, Hesari, S., & Azghandi, M. N. (2018). Maximum Power Extraction from Permanent Magnet Synchronous Generator in Wind Power Energy Systems Using Type-2 Fuzzy Logic. International Journal of Mechanical Engineering and Robotics Research7(4).)

A more complete literature coverage should include also use of Machine learning, or reinforcement learning.

(for example,  Cao, D., Hu, W., Zhao, J., Zhang, G., Zhang, B., Liu, Z., ... & Blaabjerg, F. (2020). Reinforcement learning and its applications in modern power and energy systems: A review. Journal of Modern Power Systems and Clean Energy8(6), 1029-1042.)

Also, some explanation is expected for not using capabilities such as inhibitor arcs, and colored tokens. ‏

Author Response

We would like to thank the reviewer for the valuable comments. We have carefully revised and improved the quality of the paper according to the given remarks. The changes in the manuscript are highlighted in blue. Below, we state point by point our responses to the comments, as well as the actions taken in the paper.

Comment #1: A more complete literature coverage should include also use of Timed Petri nets. (for example, Simon, D. F., Teixeira, M., & da Costa, J. P. (2022). Availability estimation in photovoltaic generation systems using Timed Petri Net simulation models. International Journal of Electrical Power & Energy Systems137, 106897.)‏
A more complete literature coverage should include also use of Fuzzy logic (for example, Hesari, S., &Azghandi, M. N. (2018). Maximum Power Extraction from Permanent Magnet Synchronous Generator in Wind Power Energy Systems Using Type-2 Fuzzy Logic. International Journal of Mechanical Engineering and Robotics Research7(4).)
A more complete literature coverage should include also use of Machine learning, or reinforcement learning.
(for example,  Cao, D., Hu, W., Zhao, J., Zhang, G., Zhang, B., Liu, Z., ... &Blaabjerg, F. (2020). Reinforcement learning and its applications in modern power and energy systems: A review. Journal of Modern Power Systems and Clean Energy8(6), 1029-1042.)
Response: Thank you for the comment. We have added and described more references in the introduction as follows:
A timed Petri net model expressing generically the networked behavior of photovoltaic systems is proposed in [15]. Resource-oriented Petri nets [16] are in turn used for scheduling, e.g., cluster tools in semiconductor manufacturing [17] or crude oil operations in refineries (hybrid coloured-timed Petri nets, being an extension of resource-oriented Petri nets) [18]. Other promising approaches to control in power energy systems use machine learning (ML) [19], deep learning (DL) [20], reinforcement learning (RL) [21] or fuzzy logic [22].
Action: The introduction has been extended and the changes in the manuscript involve lines 66-73. References [15]-[22] have been added.

Comment #2: Also, some explanation is expected for not using capabilities such as inhibitor arcs, and colored tokens.
Response: Thank you for raising this issue. The main aim of the article was to show that interpreted Petri nets (using the notions formally introduced in 2019 for specification of cyber-physical systems [23]) can be successfully used to model and analyze autonomous control components within power energy systems. The motivation of the paper was therefore to show that the application area for this type of nets is bigger than initially assumed. The interpreted Petri net type does not involve inhibitor arcs, enabling arcs or colored tokens, this is why these elements are not used in the model. Nevertheless, the usage of other Petri net types with inhibitor/enabling arcs or coloured tokens would also be interesting and can be found in the literature. In order to clear up this issue, we have added the following sentence to the manuscript:
Unlike some other types of Petri nets that might also be useful, the interpreted Petri nets do not involve inhibitor arcs, enabling arcs, or coloured tokens, which helps with the simplicity of use.
Action: The changes in the manuscript involve lines 80-82. ‏

Reviewer 3 Report

This paper uses interpreted Petri nets to describe the control logic of an autonomous component in power systems. The obtained model can describe the required behavior such that the properties, such as liveness, safeness, reachability, reversibility, and determinism, can be analyzed. This is meaningful for applications of Petri nets.

Comments:

  1. The authors present a review about the applications of Petri nets. However, it is not comprehensive. One of their applications is used as a tool for scheduling. For example, there are a lot of publications that use resource-oriented Petri nets (ROPN) as a tool for scheduling cluster tools in semiconductor manufacturing and scheduling crude oil operations in refineries. These studies make complex scheduling problems simple. Thus, it may be useful to review them.
  2. As the authors claimed, the main contribution of this paper is to use interpreted Petri nets to autonomous components in power systems. What is the challenge to do so?
  3. What is the benefit to apply Petri nets for the purpose of this paper?
  4. There is also a question whether the addressed problem can be modeled by an ordinary Petri net. If not, why? If yes, what is the advantages by using interpreted Petri nets.
  5. It is not correct to say that the Petri net shown by Fig. 1(b) is live. In fact, once t1 fires, it can never fire again, implying that it is not live. Note that p1 is a source place in the net.
  6. English needs to be polished.

Author Response

We would like to thank the reviewer for the valuable comments. We have carefully revised and improved the quality of the paper according to the given remarks. The changes in the manuscript are highlighted in blue. Below, we state point by point our responses to the comments, as well as the actions taken in the paper.

Comment #1: This paper uses interpreted Petri nets to describe the control logic of an autonomous component in power systems. The obtained model can describe the required behavior such that the properties, such as liveness, safeness, reachability, reversibility, and determinism, can be analyzed. This is meaningful for applications of Petri nets.
Response: Thank you for your positive feedback.
Action: None

Comment #2: The authors present a review about the applications of Petri nets. However, it is not comprehensive. One of their applications is used as a tool for scheduling. For example, there are a lot of publications that use resource-oriented Petri nets (ROPN) as a tool for scheduling cluster tools in semiconductor manufacturing and scheduling crude oil operations in refineries. These studies make complex scheduling problems simple. Thus, it may be useful to review them.
Response: Thank you for raising this issue. We agree that the review about the applications of Petri nets was not comprehensive. It is a hard task to include all interesting research in the paper, especially  that it also focuses on some other aspects. Following the comment we have extended the introduction by providing more references and descriptions of recent studies, including resource-oriented Petri nets. More applications have been added as follows:
A timed Petri net model expressing generically the networked behavior of photovoltaic systems is proposed in [15]. Resource-oriented Petri nets [16] are in turn used for scheduling, e.g., cluster tools in semiconductor manufacturing [17] or crude oil operations in refineries (hybrid coloured-timed Petri nets, being an extension of resource-oriented Petri nets) [18]. Other promising approaches to control in power energy systems use machine learning (ML) [19], deep learning (DL) [20], reinforcement learning (RL) [21] or fuzzy logic [22].
Action: The introduction has been extended and the changes in the manuscript involve lines 66-73. References [15]-[22] have been added.

Comment #3: As the authors claimed, the main contribution of this paper is to use interpreted Petri nets to autonomous components in power systems. What is the challenge to do so?
Response: Thank you for your question. The interpreted Petri nets, following the notions formally introduced in 2019 for specification of cyber-physical systems [23], have so-far proved to be suitable in that domain. The aim of our research was to check whether it is possible to use these types of nets also in other domains, in particular in the area of power energy systems. The results of the research show that interpreted Petri nets can be successfully used to model and analyze autonomous control components in electric power systems. There are of course some challenges in the proposed approach that have to be faced with. Firstly (already stated in the primary version of the manuscript), the knowledge of interpreted Petri net formalism is needed, as well as the possible analysis and verification methods that can be applied. This barrier can be easily overcome with some interdisciplinary research, involving both electric power system specialists as well as automatic control engineers. Secondly (not present in the primary version of the manuscript), an interpreted Petri net model is always some kind of abstraction. It means that it presents only a view on the system from some chosen perspective, and certain detailed information may intentionally be omitted at this stage of system development. Thirdly (also not present in the primary version of the manuscript), interpreted Petri nets (as a formal model) can be easily formally verified, using e.g., a model checking technique, what is not a common practice in the electric power system domain. In order to increase the quality of the manuscript, we have added more challenges to section 6.2 as follows:
Moreover, an interpreted Petri net model is always some kind of abstraction. this means that it presents only a view of the system from some chosen perspective, and certain detailed information may intentionally be omitted at this stage of system development.
and
The  challenge is here therefore to popularize formal methods in the
electric power system domain, in order to benefit from the diverse possibilities of interpreted Petri nets (as a formal model) that can be easily formally verified, using e.g., a model checking technique.
Action: The changes in the manuscript involve lines 510-512 and 519-522.

Comment #4: What is the benefit to apply Petri nets for the purpose of this paper?
Response: Thank you for this question. The benefits for applying interpreted Petri nets for the purpose of this paper are as follows. Firstly, we obtain a formal specification of an autonomous component in an electric power system (in our case it is an electric energy storage). Secondly, we can verify it (also formally), in order to check some basic properties of the model, e.g., determinism or liveness. Thirdly, the application of interpreted Petri nets gives us also the possibility to verify behavioral properties using a model checking technique or to automatically generate code in a hardware description language for a prototype implementation. In order to highlight the benefits, the following text has been added to the manuscript:
To summarize, the benefits for applying interpreted Petri nets are as follows. Firstly, an autonomous component in an electric power system is formally specified. Secondly, the prepared model can be formally verified, in order to check some basic properties, e.g., determinism or liveness. Behavioral properties can also be verified using a model checking technique. Thirdly, code in a hardware description language can be automatically generated.
Action: The changes in the manuscript involve lines 492-497.

Comment #5: There is also a question whether the addressed problem can be modeled by an ordinary Petri net. If not, why? If yes, what is the advantages by using interpreted Petri nets.
Response: Thank you for your question. The addressed problem might also be modelled either by an ordinary Petri net or by another type of Petri net. However, as this paper continues our previous research on interpreted Petri nets, we have chosen this type of Petri net as formal specification. In general, if using another type of Petri net, we would lose the information about input and output signals, so we would not be able to check their relation to each other, neither generate the prototype implementation. The advantages of using interpreted Petri nets are also discussed in our response to comment #4, and a summary of benefits has been added to the manuscript.
Action: The changes in the manuscript involve lines 492-497.

Comment #6: It is not correct to say that the Petri net shown by Fig. 1(b) is live. In fact, once t1 fires, it can never fire again, implying that it is not live. Note that p1 is a source place in the net.
Response: Thank you for pointing out this issue. We have deleted the net from Fig. 1b.
Action: Figure 1 has been changed.

Comment #7: English needs to be polished.
Response: Thank you for this comment. The manuscript has been checked by a native-speaker.
Action: The manuscript has been checked by a native-speaker.